# Can Ultrasound-Guided Femoral Vein Measurements Predict Spinal Anesthesia-Induced Hypotension in Non-Obstetric Surgery? A Prospective Observational Study

**DOI:** 10.3390/medicina58111615

**Published:** 2022-11-09

**Authors:** Ayşe Yılmaz, Ufuk Demir, Öztürk Taşkın, Veysel Garani Soylu, Zahide Doğanay

**Affiliations:** 1Department of Anesthesiology and Reanimation, Kastamonu University, 37150 Kastamonu, Turkey; 2Department of Intensive Care, 37150 Kastamonu, Turkey

**Keywords:** spinal anesthesia-induced hypotension, femoral vein diameter, femoral vein collapsibility index, non-obstetric surgery

## Abstract

*Background and objectives*: To investigate whether ultrasound (US)-guided femoral vein (FV) and inferior vena cava (IVC) measurements obtained before spinal anesthesia (SA) can be utilized to predict SA-induced hypotension (SAIH) and to identify risk factors associated with SAIH in patients undergoing non-obstetric surgery under SA. *Methods*: This was a prospective observational study conducted between November 2021 and April 2022. The study included 95 patients over the age of 18 with an American Society of Anesthesiologists (ASA) physical status score of 1 or 2. The maximum and minimum diameters of FV and IVC were measured under US guidance before SA initiation, and the collapsibility index values of FV and IVC were calculated. Patients with and without SAIH were compared. *Results*: SAIH was observed in 12 patients (12.6%). Patients with and without SAIH were similar in terms of age [58 (IQR: 19–70) vs. 48 (IQR: 21–71; *p* = 0.081) and sex (males comprised 63.9% of the SAIH and 75.0% of the non-SAIH groups) (*p* = 0.533). According to univariate analysis, no significant relationship was found between SAIH and any of the FV or IVC measurements. Multiple logistic regression analysis revealed that having an ASA class of 2 was the only independent risk factor for SAIH development (*p* = 0.014), after adjusting for age, sex, and all other relevant parameters. *Conclusions*: There is not enough evidence to accept the feasibility of utilizing US-guided FV or IVC measurements to screen for SAIH development in patients undergoing non-obstetric surgery under SA. For this, multicenter studies with more participants are needed.

## 1. Introduction

Spinal anesthesia (SA)-induced hypotension (SAIH) is one of the most common side effects of SA [1]. It is caused by the decrease in cardiac output and systemic vascular resistance after blockade of preganglionic sympathetic fibers with the administration of SA [1,2]. To prevent SAIH, prophylactic fluid administration and / or vasopressor agents may be used by some anesthesiologists [1,3]. This means taking other possible risks, such as volume overload, in order to prevent SAIH.

Although most patients without risk factors (old age, cardiac diseases, and autonomic neuropathy) tolerate SAIH well, it may increase the risk of some serious morbidities such as coronary ischemia, delirium, and even mortality [1,4,5,6]. If the likelihood of SAIH could be predicted, it would be a significant advantage for anesthesiologists. Various studies focusing on this issue have identified a number of risk factors [1,2,7,8,9]. Recently, some ultrasound (US)-guided vascular measurements, such as the inferior vena cava (IVC) collapsibility index (IVC-Ci) and IVC/Aortic diameter index [10], have been proposed by some authors as being able to predict SAIH [1,11]. However, results regarding the success of IVC measurements in predicting SAIH are conflicting [1,2,6,12]. Moreover, IVC measurements can be affected by many environmental and individual factors, including abdominal factors such as morbid obesity and intestinal gas [10]. Additionally, the IVC/Aortic diameter index is a relatively more complicated procedure and includes many of the disadvantages of IVC assessment. These problems have led researchers to seek alternative approaches to measuring blood volume status. As a result, veins that are easier to access, such as the subclavian vein, internal jugular vein, and femoral vein (FV), have been investigated, and relatively successful results have been reported [10,13,14]. Today, there are a limited number of inconsistent studies investigating the utility of FV measurements in the determination of blood volume status [10,15,16]. We found only one study [17] examining the relationship between SAIH and FV measurements. However, in this study, only the relationship between the end-expiratory transverse diameter of FV and SAIH was investigated, and the patient group consisted of only the women undergoing cesarean section under SA.

In this study, we aimed to investigate the role of some US-guided FV (primarily) and IVC measurements obtained before SA in predicting SAIH and the role of some other factors in the development of SAIH in patients undergoing non-obstetric surgery under SA.

## 2. Material and Methods

### 2.1. Ethics and Study Design

Ethical approval for this prospective observational study was acquired from the Clinical Research Ethics Committee of Kastamonu University Faculty of Medicine, Kastamonu, Turkey (No: 2020-KAEK-143-109). The study was conducted at the Department of Anesthesiology and Reanimation, Kastamonu Training and Research Hospital, from November 2021 to April 2022, according to the ethical standards stated in the Declaration of Helsinki and its amendments. All patients provided informed consent for study participation and the use of their data for scientific purposes.

### 2.2. Participants

According to descriptive statistics (effect size = 1.027) in the study by Yao S-F et al. [17] and two-sample t-test power analysis, a total sample size of 32 was found to achieve 80% power with the classical 0.05 significance level (Hintze, J. (2011). PASS 11. NCSS, LLC., Kaysville, UT, USA. https://www.ncss.com/support/faq/ (accessed on 15 October 2022)). Due to the fact that we expected a low proportion of cases with hypotension, we included more patients than calculated. 

The study included 95 patients over the age of 18 with an American Society of Anesthesiologists (ASA) physical status score of 1 or 2 [18] who were scheduled for elective operation under SA for various non-obstetric surgical indications and volunteered to participate in the study. Patients younger than 18 years of age, patients with an ASA score over 2, those with a body mass index (BMI) above 40 kg/m^2^, cases with known hypotension or hypertension history, patients with unilateral anesthetic block, subjects with hypotension detected before SA (defined as mean blood pressure < 60 mmHg and/or systolic blood pressure < 90 mmHg), pregnant women, emergency cases, cases in which SA was contraindicated, patients undergoing general anesthesia as a result of failure of SA, and individuals with diaphragmatic surgery history were excluded from the study. A total of 115 patients were evaluated for eligibility. Five patients refused to participate in the study, and 15 met the exclusion criteria (ASA score > 2 in 9 patients, emergency intervention in 3 patients, hypotension before SA in 3 patients). As a result, 95 patients completed the study.

### 2.3. Anesthesia-Related Features

#### 2.3.1. Before Spinal Anesthesia

Eight hours of fasting were planned before each procedure, and only the standard intravenous maintenance fluid regimen (Ringer lactate, 4 mL/kg/h for the first 10 kg, 2 mL/kg/h for the second 10 kg, and 1 mL/kg/h for the remaining body weight) [19] was administered during this period. No fluid preload was applied to the patients after they were transferred to the operating room. The patients were monitored in the supine position for continuous electrocardiograms, non-invasive blood pressure, and oxygen saturation measurements. The patients’ age, sex, BMI (kg/m^2^), ASA scores, and the department that scheduled the surgery were recorded. The baseline values of systolic blood pressure (SBP) (mmHg), diastolic blood pressure (mmHg) (DBP), heart rate (beats/min) were measured and recorded. The mean blood pressure (MBP) (mmHg) was calculated according to SBP and DBP with the following formula: MBP = DBP + [(SBP − DBP)/3] [20]. Necessary US assessments were performed, and all results were recorded.

#### 2.3.2. During and after Spinal Anesthesia

Participants underwent a standard SA procedure. Ringer lactate solution infusion was started at 10 mL/kg/h through an 18-gauge intravenous line following SA initiation. Pharmacological premedication was not applied. Spinal anesthesia was administered to all patients in the sitting position through the L3–4 and L4–5 intervals. In each patient, 2.5 mg of heavy bupivacaine was applied to the spinal space through a 26-gauge spinal needle. No opioids were used in spinal anesthesia. The patients were placed in the supine position after spinal anesthesia and remained in the supine position during evaluations. SBP, DBP, MBP, and heart rate values were recorded at 5 time points (at initiation [0 min], 1st minute, the 5th minute, the 10th minute, and the 15th minute of SA administration). Fifteen minutes after SA administration, the sensory block was checked with a pinprick test before starting the surgery. Sensory block in all patients was at the T8–10 level.

### 2.4. Sonographic Technique and Measurements

All US procedures were performed by the same anesthesiologist with a Siemens Acuson P500 device (Siemens, Mountain View, CA, USA) before starting the SA process and after baseline blood pressure and heart rate were measured.

#### 2.4.1. Inferior Vena Cava Measurements

The US examinations of the IVC were performed as described by the American Society of Echocardiography [21] with B (brightness) and M (motion) imaging modes in a longitudinal section by using a curvilinear probe [(CH5-2); 1.4–5.0 MHz, Siemens, USA] with the patients in the supine position. First, the probe was placed in the sub-xiphoid region, and the right atrium was imaged on B-mode. The measurements were taken from the best view of the IVC, 2–4 cm below the right atrium [22]. In this image, a phasic collapse of the IVC Doppler waveform with compression and respiration was observed. The maximum (IVCD_max_) and minimum (IVCD_min_) anterior and posterior diameters of the IVC, respectively, during expiration and inspiration, were measured on M-mode imaging, and images were saved for reevaluation after the procedure. Using the obtained values, the IVC-Ci was calculated with the following formula: IVC-Ci = [(IVCD_max_ − IVCD_min_)/(IVCD_max_)] × 100 [22].

#### 2.4.2. Femoral Vein Measurements

For the purpose of standardization, right femoral vein measurements were used in the study. The FV was visualized with B-mode US, 2–5 cm below the level of the inguinal ligament, just above the inguinal canal, from where the femoral artery was best palpated, without applying any pressure that could affect the diameter of the FV [10]. Next, FV diameter variation during the respiratory cycle [maximum FV diameter (FVD_max_), minimum FV diameter (FVD_min_)] was measured in the transverse view using a high-frequency linear array transducer [(VF 13–5; 4.1–12.1 MHz), Siemens, USA] on M-mode, and images were saved for reevaluation after the procedure. Using the values obtained, the FV collapsibility index (FV-Ci) was calculated with the following formula: FV-Ci = [(FVD_max_–FVD_min_)/(FVD_max_)] × 100 [10] (Figure 1).

### 2.5. SAIH Criteria and Treatment

The criteria of SAIH were considered MBP below 60 mmHg and/or SBP below 90 mmHg, and/or SBP decreasing by 30% compared to the baseline value, in measurements made within 15 min after SA administration [1].

Patients who developed hypotension were administered intravenous fluid (250 mL of crystalloid infusion in 10 min) and a vasopressor agent (5 mg of ephedrine at 2 min intervals) until a 20% increase in SBP or MBP was observed [23,24]. In the meantime, patients were followed up for possible complications such as nausea, vomiting, allergies, etc., and managed as per standard protocol.

### 2.6. Outcomes

The primary outcome was to evaluate whether FV-Ci in particular (and/or IVC-Ci) could predict SAIH. The secondary outcome measure was defined as identifying other possible risk factors associated with SAIH development.

### 2.7. Statistical Analysis

All analyses were performed on IBM SPSS for Windows, Version 25.0 (IBM Corp., Armonk, NY, USA). For the normality check, the Shapiro–Wilk test was used. Data are given as the median (minimum–maximum) for continuous variables due to the non-normality of their distribution in analyzed variables, while frequency (percentage) is used for categorical variables. Repeated measurements were analyzed with Friedman’s analysis of variances by ranks. Pairwise comparisons were adjusted by the Bonferroni correction method. Pairwise comparison results are represented by letter notation. The same letters denote the lack of a statistically significant difference between the respective groups. For example, if both measurements are indicated with “a”, there is no statistically significant difference between these groups, whereas groups without the same letter demonstrate significant differences. Between-group comparisons of continuous variables were performed with the Mann–Whitney U test. Between-group comparisons of categorical variables were performed with appropriate chi-square tests, the Fisher’s exact test, or the Fisher-Freeman-Halton test. Multiple logistic regression analysis was performed to determine independent risk factors associated with SAIH. *p* < 0.05 values were accepted as statistically significant results.

## 3. Results

Sixty-two male (65.3%) and 33 female patients were included in the study, and the median age of the patients was 51 (min–max: 19–71) years. SAIH was diagnosed in 12 patients (12.6%). The median age of the patients with SAIH was 58 (min–max: 19–70) years, while patients without SAIH had a median age of 48 (min–max: 21–71) years. All patients’ characteristics, US measurements, and univariate analysis results showing their relationship with hypotension are summarized in Table 1. 

There was no significant difference in age between the two groups (*p* = 0.081). Males comprised 63.9% of the patients with SAIH and 75.0% of those without SAIH (*p* = 0.533). According to univariate analysis results, no significant relationship was found between SAIH and any of the FV or IVC measurements. There was a significant relationship between SAIH and having an ASA score of 2 (*p* = 0.002). A total of 5 mg of ephedrine at 2-min intervals was administered to only one of the patients with SAIH, and fluid boluses were sufficient in the other 11 patients. Due to the low number of patients in need of vasopressors, a correlation study could not be performed between the measured values and the total dose of vasopressors.

It was observed that SBP, DBP, MBP, and heart rate decreased significantly towards the 10th and 15th minutes following SA (*p* < 0.001 for all) (Table 2).

Multiple logistic regression analysis revealed that ASA classification was the only factor that was independently associated with SAIH. Patients with ASA 2 had a 14.982-fold greater risk of SAIH compared to those with ASA 1, after adjusting for other variables (OR: 14.982; 95% CI: 1.727–129.930; *p* = 0.014) (Table 3).

## 4. Discussion

Since it can cause serious adverse consequences, SAIH is a complication that should be considered carefully, especially in patients with hemodynamic risk factors [4,5]. To the best of our knowledge, this is the first study to investigate the role of FV diameters and FV-Ci in predicting patients’ development of SAIH during non-obstetric surgery. It was observed that SAIH risk was not associated with FV diameters or FV-Ci. We also did not find any significant relationship between SAIH development and IVC diameters or IVC-Ci. After adjusting for all relevant factors, only having an ASA score of 2 (compared to ASA 1) was found to be an independent risk factor for SAIH.

There are many different studies reporting results on this subject. Sumit R. Chowdhury et al. [25] included 50 patients, and 34% developed SAIH. Shayak Roy et al. [6] reported that 19.37% of 129 patients developed SAIH. Bernd Hartmann et al. [8] assessed data sets from 3315 patients receiving SA from 1 January 1997, to 5 August 2000, by using the automatic anesthesia record-keeping system NarkoData. Hypotension meeting the predefined criteria occurred in 166 (5.4%) patients. Although our study seems to have a lower SAIH rate compared to these first two studies, many factors, such as the preoperative intravascular volume status of the patients, additional disease, age, and the amount of spinal anesthetic agent used, are effective in this regard. We believe that such a result will contribute to the literature.

Prophylactic administration of volume or vasopressor agents to prevent SAIH can increase the risk of additional morbidities, including pulmonary edema, congestive cardiac failure, and renal dysfunction, especially in patients with cardiac risk factors [6,24]. Instead of using a one-size-fits-all management approach, using prophylactic treatment only for patients with a high risk of developing SAIH is highly preferable [1]. However, detecting these patients necessitates the identification of noninvasive, easy-to-apply, low-cost screening tools. To this end, in this study, we used US-guided FV measurements, which had been tested in previous studies but were found to have inconsistent capabilities. Our results show that FV measurements are not useful tools for estimating SAIH. Yao et al. investigated the end-expiratory transverse diameter of the right common FV to assess its potential in predicting the occurrence of SAIH in women undergoing cesarean section. They concluded that the transverse diameter of the right common FV in the hypotension group was significantly larger than in the group without hypotension and that the transverse diameter of the right common FV measured by the US was associated with the occurrence of SAIH [17]. In the study of Kent et al., it was stated that the use of FV-Ci as a primary blood volume status assessment tool for clinical decision support was not reliable [10]. Conversely, another study found the FV diameter accuracy measured by US to predict central venous pressure comparable to that of IVC diameter, suggesting that FV diameter may provide an alternative approach when IVC imaging is difficult [15], similar to the results reported by Zidan and colleagues [26]. FV assessment may have some evaluation advantages over IVC because it is a more superficial vessel, lacks imaging difficulties caused by abdominal factors, and allows measurement that is minimally affected by variations in the respiratory cycle [10,15,16,27]. However, FV assessment also has disadvantages compared to IVC. For example, because the FV is a more superficial vein, measurements are more likely to be affected by the pressure of the US probe during application [10]. Although US evaluations were made by the same radiologist in this study, the results may have been affected due to the difficulty of standardizing the pressure during measurement. Another possible problem is that the FV is a smaller vessel compared to the IVC; thus, it can be expected that the obtained values will be affected to a greater degree by any factors that negatively affect US measurement. The low number of patients with SAIH may be another factor limiting our analyses. The identification of these problems may have value for future studies on the subject and demonstrate the need for more comprehensive studies evaluating relationships between FV measurements and SAIH.

Measurements of the IVC have been shown to reflect intravascular volume [10]. Based on this feature of the IVC, many researchers investigated the role of US-guided IVC measurements in predicting SAIH, and much of this research showed inconsistent results [1,2,6,22]. In this study, we did not find IVC diameters or IVC-Ci to be a predictor of SAIH. In a prospective, randomized cohort study, prophylactic hypotension treatment (crystalloid fluid administration) was administered to a group of patients with an IVC-Ci of >36% (measured before SA), while others did not receive this treatment. The incidence of SAIH and the need for fluid and vasopressor agents after SA were found to be significantly lower in the prophylactic treatment group compared to the untreated group. However, this study did not show a significant relationship between IVC-Ci and the decrease in MBP [2]. In other similar studies, it was shown that US-guided IVC measurements may help reduce the incidence of SAIH [22,24,28]. While some researchers claimed that IVC-Ci had significant sensitivity and specificity in predicting SAIH [1,22], others did not find any notable relationship between IVC-Ci and SAIH [2,6,28]. Blood volume, cardiac output, and vascular resistance are the main factors influencing blood pressure [20]. Therefore, it can be expected that an underlying factor affecting these parameters could also affect blood pressure. The IVC is a high-capacitance vessel that is sensitive to volume changes and reflects blood volume status. Intravascular volume changes in the respiratory cycle can be affected by many factors, such as age, anthropometric measurements, asthma, respiratory tract infection, chronic obstructive diseases, and intrathoracic and intra-abdominal pressures [1,6,29,30]. All of these may be possible reasons for inconsistent results in studies and could also explain our findings.

In order to contribute to future comprehensive studies on this topic, we also investigated whether other risk factors could be identified. As a result of the multiple logistic regression analysis, only an ASA score of 2 (compared to an ASA score of 1) was found to be an independent risk factor for SAIH. Various studies focusing on this issue have identified a number of risk factors for the development of SAIH. The most prominent are as follows: higher sensory block level and older age (as two fundamental risk factors), emergency surgery, presence of hypertension, chronic alcohol consumption, baseline SBP greater than 120 mmHg, combination with general anesthesia, and spinal puncture at or above L2–L3 [1,2,7,8,9,31]. In a prospective observational study, it was stated that there was no significant relationship between ASA and SAIH, contrary to the results of the current study [1]. In another study’s univariate analysis, a significant correlation was found between having an ASA score of 2 and low baseline SBP, DBP, and MBP and SAIH development. However, none of these were identified as significant risk factors in multivariable analysis [6]. It has been shown that SAIH susceptibility increases as the ASA risk category increases, even in patients who are pre-hydrated before anesthesia [32]. Although current evidence appears to be very limited, there is a need for a SAIH risk classification that can enable the identification of high-risk patients before SA administration, thereby easing clinical decision-making.

The present research has two important advantages. First, it is the first study to evaluate pre-anesthetic USG-guided FV measurements to predict SAIH development in patients undergoing non-obstetric surgery. Secondly, not only the collapsibility index of the veins but also the relationship between their diameters and SAIH were investigated. Some limitations of the study should also be noted. It is a single-center study, so the ability to generalize the results is limited. Only measurements at specified minutes were taken into account; and continuous measurements were not possible, which may have caused under-diagnosis of SAIH. Although patients were given only maintenance fluid before anesthesia, this approach also has the potential to affect the likelihood of hypotension. Only patients with an ASA risk score of 1–2 were included, and therefore, the results only cover patients in this risk group and cannot be generalized to other ASA groups. Studies with larger patient groups are needed to determine whether the IVC and FM indices are predictive in this context.

In conclusion, we did not find that FV diameters, FV-Ci, IVC diameters, and IVC-Ci were parameters that could predict SAIH in patients undergoing non-obstetric surgery under SA. After adjusting for all relevant parameters, only an ASA score of 2 (compared to ASA 1) was identified as an independent risk factor for SAIH in this patient group. The current evidence is insufficient to make a definitive conclusion; however, it seems that US-guided FV or IVC measurements cannot be reliably used to assess SAIH likelihood in patients undergoing non-obstetric surgery under SA.

## Figures and Tables

**Figure 1 medicina-58-01615-f001:**
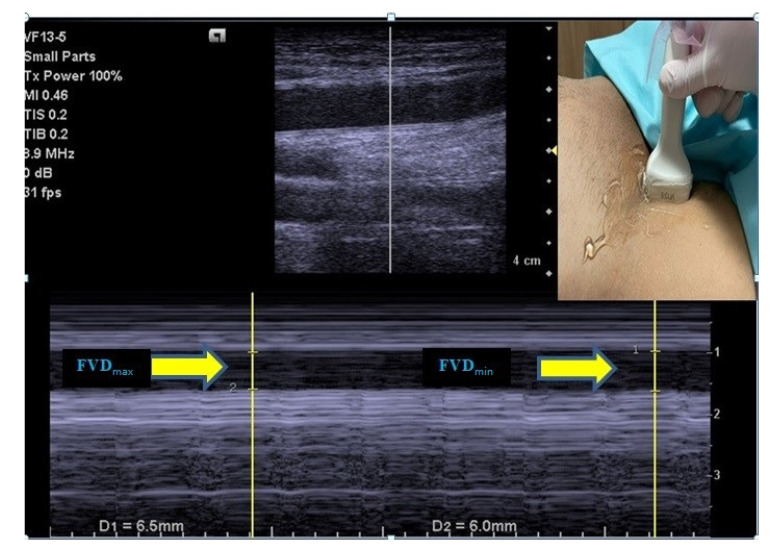
Maximum and minimum diameter measurements of the femoral vein.

**Table 1 medicina-58-01615-t001:** Summary of patient characteristics and their relationship with hypotension.

		Hypotension		
	All Patients	No (n = 83)	Yes (n = 12)	*p*
Age	51 (19–71)	48 (21–71)	58 (19–70)	0.081
Sex				
Male	62 (65.3%)	53 (63.9%)	9 (75.0%)	0.533
Female	33 (34.7%)	30 (36.1%)	3 (25.0%)	
Body mass index (kg/m^2^)	26.0 (16.5–39.0)	26.1 (16.5–35.0)	25.4 (19.0–39.0)	0.494
ASA classification				
1	51 (53.7%)	50 (60.2%)	1 (8.3%)	0.002
2	44 (46.3%)	33 (39.8%)	11 (91.7%)	
Department of surgery				
Urology	19 (20.0%)	16 (19.3%)	3 (25.0%)	0.240
Orthopedics	38 (40.0%)	36 (43.4%)	2 (16.7%)	
Cardiovascular	13 (13.7%)	10 (12.0%)	3 (25.0%)	
General	25 (26.3%)	21 (25.3%)	4 (33.3%)	
Inferior vena cava				
Maximum diameter (cm)	2.45 (1.30–3.20)	2.56 (1.30–3.20)	2.05 (1.89–2.87)	0.728
Minimum diameter (cm)	2.01 (0.70–2.90)	2.04 (0.70–2.90)	1.96 (1.34–2.30)	0.556
Collapsibility index (%)	18.52 (1.99–58.58)	18.29 (4.49–58.58)	20.00 (1.99–31.98)	0.556
Femoral vein				
Maximum diameter (cm)	0.93 (0.57–1.40)	0.93 (0.59–1.40)	0.95 (0.57–1.19)	0.982
Minimum diameter (cm)	0.78 (0.39–1.03)	0.78 (0.42–1.03)	0.83 (0.39–1.02)	0.766
Collapsibility index (%)	15.52 (1.32–40.71)	16.05 (1.32–40.71)	10.99 (4.41–34.45)	0.370

Data are given as the median (minimum–maximum) for continuous variables due to the non-normality of their distribution and as the frequency (percentage) for categorical variables. Abbreviations; ASA: American Society of Anesthesiologists.

**Table 2 medicina-58-01615-t002:** Summary of other measurements performed in the study.

Systolic blood pressure (mmHg)	
Baseline	146 (107–178) ^a^
0th minute	142 (110–171) ^a^
1st minute	131 (95–164) ^b^
5th minute	124 (90–168) ^c^
10th minute	118 (73–157) ^d^
15th minute	118 (77–160) ^d^
*p*	<0.001
Diastolic blood pressure (mmHg)	
Baseline	82 (58–104) ^a^
0th minute	75 (54–103) ^a^
1st minute	70 (48–99) ^b^
5th minute	67 (45–96) ^bc^
10th minute	63 (42–92) ^d^
15th minute	64 (51–98) ^cd^
*p*	<0.001
Mean blood pressure (mmHg)	
Baseline	103 (61–129) ^a^
0th minute	101 (66–127) ^a^
1st minute	94 (61–121) ^b^
5th minute	89 (63–116) ^b^
10th minute	81 (61–114) ^c^
15th minute	82 (61–117) ^c^
*p*	<0.001
Heart rate (beats/min)	
Baseline	76 (58–118) ^a^
0th minute	76 (51–117) ^a^
1st minute	78 (55–114) ^a^
5th minute	76 (53–113) ^ab^
10th minute	76 (46–108) ^b^
15th minute	73 (51–107) ^b^
*p*	<0.001
Hypotension criteria met	
Decrease in systolic blood pressure, ≥30%	12 (12.6%)
Systolic blood pressure, <90	5 (5.3%)
Mean blood pressure, <60	0 (0.0%)

Data are given as the median (minimum–maximum) for continuous variables due to the non-normality of their distribution and as the frequency (percentage) for categorical variables. ^a,b,c,d^: Same letters denote the lack of statistically significant difference between repeated measurements.

**Table 3 medicina-58-01615-t003:** Significant factors independently associated with hypotension, multiple logistic regression analysis.

	β Coefficient	Standard Error	*p*	Exp(β)	95.0% CI for Exp(β)	
Age	0.051	0.032	0.114	1.052	0.988	1.121
Sex, Female	−0.839	0.808	0.299	0.432	0.089	2.105
Body mass index	−0.028	0.099	0.779	0.973	0.801	1.181
ASA classification, 2	2.707	1.102	0.014	14.982	1.727	129.930
Department, Orthopedics	−1.500	0.946	0.113	0.223	0.035	1.426
Inferior vena cava collapsibility index	−0.049	0.042	0.235	0.952	0.877	1.033
Femoral vein collapsibility index	−0.004	0.037	0.925	0.996	0.926	1.072
Constant	−4.092	3.270	0.211	0.017		

CI: Confidence Interval, Nagelkerke R^2^ = 0.366; Abbreviations; ASA: American Society of Anesthesiologists.

## Data Availability

Not applicable.

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
