# Peer review of "Can Ultrasound-Guided Femoral Vein Measurements Predict Spinal Anesthesia-Induced Hypotension in Non-Obstetric Surgery? A Prospective Observational Study"

_medicina, 2022, doi:10.3390/medicina58111615_

Round 1

Reviewer 1 Report (Previous Reviewer 4)

Accept revision

Author Response

Response to Reviewer

First of all, thank you very much for your kind evaluation.

Reviewer 2 Report (Previous Reviewer 1)

Thank you for your improvements.

Your response 1 needs to be included in the manuscript.

Author Response

Response to Reviewer

First of all, thank you very much for your kind evaluation.

  Point: Added as suggested.

This manuscript is a resubmission of an earlier submission. The following is a list of the peer review reports and author responses from that submission.

Round 1

Reviewer 1 Report

The authors performed a prospective observational study to investigate wether SIAH can be predicted by dynamic measurements of IVC an FM. The investigated parameters did not predict SIAH in the study.

I do agree with the authors, that these parameters probable are not suitable for predicting SIAH.

The incidence of SAIH (12/95) was much lower than the cited study European journal of  anaesthesiology. 2019;36(4):297-302. 282 (Incidence of SAIH 45% (45/100)).

The sample size needed to provide enough statistical power for a negative result was not calculated.

The sample size is not big enough to prove that IVC and FM indices are not predictive

Studies with negative results merit publication only if the statistical power is adequate and results are clinically relevant.

Reviewer 2 Report

Thank you for the opportunity to review an interesting study.

  1. SAIH  is developed by spinal anesthesia, but there was no detail of spinal anesthesia in your study. used local anesthetics and dose, whether opioids are used or not, and level of anesthesia. They can affect developing SAIH and can improve your study.

  2. Was there premedication?

  3. How many patients were eligible for your study? How many patients were excluded and what were the reasons?

  4. Can you show measurement methods of inferior vena cava and femoral vein with images of ultrasound?

  5. In table 1, What do the superscripts a and b mean?

  6. In Table 2, age, gender, BMI, ASA-PS and surgery have the same contents as in table 1, and are only classified according to hypotension. It may be better merging table 1 and table 2 while comparing between hypotension and non-hypotension, or dividing patients' characteristics and measurements as each table while comparing between hypotension and non-hypotension.

Reviewer 3 Report

I read with great interest the article titled "Can ultrasound-guided femoral vein measurements predict spinal anesthesia-induced hypotension in non-obstetric surgery? A prospective observational study." It is agreeable that if the likelihood of post spinal hypotension can be predicted, it would be of a great benefit for all anesthesiologists. 

Comments and suggestions: 

Please follow the journal's instruction for authors, manuscript template, and referencing style. 

"Fifteen minutes after SA administration, sensory block was checked with pinprick test before starting the surgery" Did you document the level of block? As the level of block can significantly affect the rate of post spinal hypotension. 

Did you document the medications given for spinal anesthesia and doses of these medications?

Were the femoral vein measurements conducted on the right or left femoral vein? 

Please mention the full terms upon it's appearance in the manuscript, before using the abbreviations. 

Did you perform power analysis for the sample size of the current study? Is it a representative sample size?

Table 1:

"Same letters denote the lack of statistically significant difference between repeated measurements."

Explain this for the readers in the statistical analysis section. 

Table 3:

Department of surgery: you included four departments, use one as  a reference standard for all comparisons, and then each of the departments included will have its own OR. 

Did you document the overall dose of vasopressors? You may find significant correlations between the investigated measurements and index and the overall dose of vasopressors. 

In the conclusion paragraph, you mentioned that "In conclusion, FV diameters, FV-Ci, IVC diameters and IVC-Ci were not found to be suitable tools to predict SAIH in patients undergoing non-obstetric surgery under SA."

Please correct the statement, as you can't generalize that they are not suitable, but you can indicate that the current study did not find significant results. 

Reviewer 4 Report

Excellent well written prospective observational study.

Nice discussion and literature review.

As the main question that I have is that the authors found no difference between groups, what is the power of the study to be sure that there is not a type 2 error. The finding of no difference, but that the sample size is not large enough. 

Unless I have missed something, I could not find their power statement in the manuscript.